# Potential Risks of PM_2.5_-Bound Polycyclic Aromatic Hydrocarbons and Heavy Metals from Inland and Marine Directions for a Marine Background Site in North China

**DOI:** 10.3390/toxics10010032

**Published:** 2022-01-11

**Authors:** Qianqian Xue, Yingze Tian, Xinyi Liu, Xiaojun Wang, Bo Huang, Hongxia Zhu, Yinchang Feng

**Affiliations:** 1The State Environmental Protection Key Laboratory of Urban Air Particulate Matter Pollution Prevention and Control, College of Environmental Science and Engineering, Nankai University, Tianjin 300350, China; xueqq@mail.nankai.edu.cn (Q.X.); liuxinyi@mail.nankai.edu.cn (X.L.); 2CMA-NKU Cooperative Laboratory for Atmospheric Environment-Health Research (CLAER/CMA-NKU), Tianjin 300350, China; 3Yantai Environmental Monitoring Centre, Yantai 264003, China; WangXj121@hotmail.com; 4Guangzhou Hexin Instrument Co., Ltd., Guangzhou 510530, China; b.huang@hxmass.com; 5China National Environmental Monitoring Center, Beijing 100012, China; zhuhx@cnemc.cn

**Keywords:** Tuoji Island, organic/inorganic tracers, toxic potency, long-distance transport, risk assessment

## Abstract

Ambient PM_2.5_-bound ions, OC, EC, heavy metals (HMs), 18 polycyclic aromatic hydrocarbons (PAHs), 7 hopanes, and 29 n-alkanes were detected at Tuoji Island (TI), the only marine background atmospheric monitoring station in North China. The annual PM_2.5_ average concentration was 47 ± 31 μg m^−3^, and the average concentrations of the compositions in PM_2.5_ were higher in cold seasons than in warm seasons. The cancer and non-cancer risks of HMs and PAHs in cold seasons were also higher than in warm seasons. BaP, Ni, and As dominated the ∑HQ (hazard quotient) in cold seasons, while the non-carcinogenic risk in warm seasons was mainly dominated by Ni, Mn, and As. The ILCR (incremental lifetime cancer risk) values associated with Cr and As were higher in the cold season, while ILCR-Ni values were higher in the warm season. The backward trajectory was calculated to identify the potential directions of air mass at TI. Through the diagnostic ratios of organic and inorganic tracers, the sources of particulate matter in different directions were judged. It was found that ship emissions and sea salt were the main sources from marine directions, while coal combustion, vehicles emissions, industrial process, and secondary aerosols were the main source categories for inland directions. In addition, potential HM and PAH risks from inland and marine directions were explored. The non-cancerous effects of TI were mainly affected by inland transport, especially from the southeast, northwest, and west-northwest. The cancerous effects of TI were mainly simultaneously affected by the inland direction and marine direction of transport.

## 1. Introduction

Airborne fine particulate matter (PM_2.5_) is of significant interest owing to its negative effects on climate change and human health all over the world [1,2,3,4,5,6,7]. Studies of compositions of PMs at background sites can provide valuable information on the impacts of anthropogenic activities and the influence of long-distance transmission [8,9,10]. Tuoji Island (TI) is the only marine national background atmospheric monitoring station used for regional ambient air quality monitoring in North China [9,10,11]. It gives us a special and rare opportunity to study the influence of inland and marine transmission and some special sources (such as ship emissions) for the marine background atmospheric monitoring station.

Research has shown that shipping emissions have a significant impact on air pollution in coastal areas and ports [12,13,14]. Approximately sixty thousand deaths from cardiovascular disease and lung cancer are related to ship emissions [12,13,15]. So far, most studies conducted on ship emissions have focused on the concentration and risk assessment of heavy metals [12,13,15]. However, organic compounds, such as PAHs, also constitute a significant class of matter emitted from ships [16,17]. As the only marine national background atmospheric monitoring station, there is almost no research on how ship emissions affect health risks associated with PMs in the background site at TI [12].

There is a wide variety of components in particles, some of which can cause adverse effects in the human body, such as heavy metals (HMs), polycyclic aromatic hydrocarbons (PAHs), etc. [4,17,18,19,20,21]. Some components can be used as markers to identify the sources, such as HMs, PAHs, hopanes, and n-alkanes [22,23,24,25]. They can derive from anthropogenic activities (such as biomass burning, coal combustion, oil combustion, industrial processes, etc.) and natural sources (such as > C29 odd n-alkanes from vegetation leaf surfaces) [21,26,27,28]. Some scholars have conducted related studies on source markers and the risk assessment of particulate matter (PM) in urban cities [1,2,3,4,22,23,24,28,29,30], but those on background sites are limited [9,10,11]. A few studies have studied the atmospheric PMs at TI, mainly examining the compositions of PMs, PM source apportionment, PAH source apportionment, and PAH risk assessment [9,10,11,16,31,32]. These studies have proved that polluted air mainly comes from outside transportation, such as the North China Plain and Liaoning [9]. A high toxic potency of PAHs in the air was determined in middle Liaoning and south Shandong Peninsula [11]. However, there are few studies that compare HMs and PAHs and comprehensively assess the health risks from inland and marine directions at the marine background site by combining a variety of organic markers, HMs, PAHs. and air mass trajectories [9,10,11,16,31,32].

In view of the above shortcomings, the study mainly aimed to achieve the following three goals: (1) to characterize the composition of PM_2.5_ by analyzing organic carbon (OC), elemental carbon (EC), elements, water-soluble ions, 18 PAHs, 7 hopanes, and 29 n-alkanes; (2) to use trajectory directions and inorganic/organic markers to distinguish main source categories from inland and marine directions; and (3) to estimate the health risk of PM_2.5_-bound PAHs and HMs from inland and marine transmission at the background site. The synchronous measurement of inorganic and organic components of PM_2.5_ at TI is uncommon, and there is a lack of organic markers to identify the sources of PM_2.5_ from inland and marine directions at TI. This provided us with a specific opportunity for a contrastive study of PM_2.5_ composition and the health risks posed by HMs and PAHs from inland and marine directions.

## 2. Materials and Methods

### 2.1. Sampling Site and Sample Collection

#### 2.1.1. Sampling Site

The sampling site (120.75° E, 38.19° N) at Tuoji Island was located in the middle of the Bohai Sea, about 40 km north of the Penglai, Shandong Peninsula (Appendix A). This site is the only marine atmospheric background station in northern China [11]. There is almost no industrial activity on the island, which covers an area of 7.1 km^2^ and has a permanent population of 3000. There are several small villages, and the main industry is fishery. At this site, the airflow is not obstructed and is affected by air masses from different directions [10].

#### 2.1.2. PM_2.5_ Sample Collection

PM_2.5_ samples were collected from 1 January to 30 October in 2017, covering the cold season (January and March, 32 PM_2.5_ samples) and the warm season (August and October, 23 PM_2.5_ samples), and each sample was collected for 24 h. In northern China, March belongs to the end of winter and the beginning of spring. Fifty-five samples were collected by a mid-volume (100 L/min) suspended particle sampler (TH-150C; Wuhan Tianhong, China). Quartz and polypropylene fiber filters with 90 mm diameter were used to collect samples (PALLFLEX, Pall Corporation, NY, USA). Details of quality control and quality assurance of sampling has been described in our previous reports [4,16,33,34].

### 2.2. Sample Chemical Analysis

#### 2.2.1. Conventional Components Analysis

Ions (Cl^−^, SO_4_^2−^, NO_3_^−^, NH_4_^+^, etc.) were analyzed using an ion chromatograph (IC) system (ICS-900; DIONEX, Sunnyvale, CA, USA). Firstly, a quarter of quartz filters were cut into small pieces, placed directly into tubes and ultrasonically extracted with 8 mL deionized water for 20 min. After extraction, the solution was stored in a refrigerator for 24 h. The supernatant was aspirated with a needle tube and injected into the vial through two 0.22 µm filters. Finally, ion chromatography was used to analyze the solution to quantitative ions [4,15,16,17,30,35]. Seventeen inorganic elements (including Na, Mg, Al, Si, K, Cr, Co, Ni, As, Cd, and Pb) were analyzed by an inductively coupled plasma atomic emission spectrometer (iCAP 7400 ICP-AES; Thermo Fisher Scientific, Waltham, MA, USA). Firstly, one-eighth filters were treated with 10 mL mixed digestion solution (2 mL nitric acid, 6 mL hydrochloric acid, 2 mL hydrogen peroxide). They were covered and put into the microwave turntable (MARS; CEM Corporation, NC, USA). The heating procedure was as follows: raise to 120 °C for 8 min; raise to 150 °C for 8 min; after 3 min, raise to 180 °C for 8 min; after 3 min, raise to 200 °C for 10 min. Then, the solutions were cooled and filtered through a 0.22 μm microporous filter. The digested samples were subsequently atomized in the atomizer of a plasma emission spectrometer. Lastly, the target element was vaporized, ionized, excited, and radiated out of the characteristic line in the plasma torch. Within a certain concentration range, the characteristic line intensity is proportional to the element concentration [4,15,16,17,30,35]. A thermal/optical carbon aerosol analyzer (DRI model 2001A; Desert Research Institute, Reno, NV, USA) was used to analyze organic carbon (OC) and elemental carbon (EC). OC and EC were analyzed based on a punch with a quartz filter of 0.588 cm^2^. The thermal/optical carbon aerosol analyzer detected OC1, OC2, OC3, and OC4 in a pure helium atmosphere at temperatures of 140, 280, 480, and 580 °C, respectively. Similarly, the oven temperature was increased to 540, 780, and 840 °C for EC1, EC2, and EC3 analyses, respectively, in a 2% O_2_ atmosphere. Organic pyrolysed carbon (OPC) was also detected after adding oxygen. Finally, the OC and EC concentrations were calculated by OC = OC1 + OC2 + OC3 + OC4 + OPC and EC = EC1 + EC2 + EC3 − OPC., respectively [4,15,16,17,30,35].

#### 2.2.2. Organic Components Analysis

Eighteen PAHs, seven hopanes, two steranes, and n-alkanes (C7-C40) were determined using a gas chromatography/mass spectrometer (GC/MS, Agilent 7890B/5977B, CA, USA). Detailed information about the compounds for conventional and organic components is given in Appendix A. Further information on our chemical analysis of organic compounds followed in this study is reported elsewhere [23,24]. In brief, the GC/MS was equipped with a DB-5MS fused-silica capillary column (30 m × 0.25 mm, 0.25 mm film thickness, Agilent Technology, Santa Clara, CA, USA) and 1 mL injected into a splitless injector. Pure helium (purity of 99.9995%) was used as a carrier gas at a constant flow rate of 1.0 mL min^−1^. To analyze PAHs, hopanes, and steranes, inlet and temperament transmission line temperatures were set to 230 °C and 280 °C, and the initial ionization temperature was set to 280 °C. The chromatography temperature program was set as follows: the column oven temperature was initially held at 50 °C for 2 min, increased to 240 °C at a rate of 10 °C min^−1^, and finally increased to 280 °C at 5 °C min^−1^ and held for 20 min. Mass spectrometry conditions was set as follows: the EI mode was selected and the ionization energy level was 70 eV. An internal standard (IS) (naphthalene-D8, acenaphthene-D10, phenanthrene-D10, phenanthrene-D12, pyrene-D10, hexamethylbenzene) was applied to the samples to qualify actual volumes of the PAHs and hopanes. As for n-alkane analysis conditions, inlet and temperament transmission line temperatures were 300 °C, and the initial ionization temperature was 300 °C. The chromatography temperature program consisted of the following steps: column oven temperature was initially held at 50 °C, increased to 60 °C at a rate of 2.5 °C min^−1^, then increased to 280 °C at a rate of 22 °C min^−1^, and finally increased to 325 °C at 30 °C min^−1^ and held for 11 min.

### 2.3. Quality Assurance and Quality Control

During the experiment, strict quality control procedures were implemented. In order to control data quality, method/lab blanks and parallel samples were used when analyzing each batch of samples. In this study, the correlation coefficient (r) of the standard curve exceeded 0.99. Details about the analytical recovery of PAHs, hopanes, and n-alkanes are presented in Appendix A; most compounds were recovered with a recovery efficiency of 80–120% except for individual species. The reported value has been corrected for the blank or recovery rate [2]. The limit of method detection (LOD) and limit of quantitation (LOQ) for conventional and organic components are given in Appendix A. In this part, some measurement uncertainties may occur.

### 2.4. Health Risk Assessment for Different Directions

#### 2.4.1. Step 1: Back-Trajectory Calculation and Cluster Analysis

The back trajectory model is a widely used and vital tool to identify the potential directions of air parcel sources [9,10,30,36,37,38,39,40,41]. Generally, trajectories show the paths of air mass [38,39,40,41], and grouping back trajectories arrive at a receptor site during the sampling period. The back trajectories indicate where the potential sources of PAHs and harmful heavy metals (HMs) in PM_2.5_ originated. Backward trajectories of 72 h duration were calculated every 6 h at 500 m above ground level. A total of 493 trajectories were generated through the clustering function in the Hybrid Single Particle Lagrangian Integrated Trajectory (HYSPLIT) model [38,39,40,41]. To identify the potential directions of air mass and the effect of different transport routes on particle chemical compositions in TI, the backward trajectory was calculated during the sampling period. Five groups air masses were identified according to their transport directions and area of travel. Therefore, in this step, the direction and the backward trajectory concentrated in each direction were obtained.

#### 2.4.2. Step 2: Source Identification for Different Directions

To explore the sources for different directions, tracers and diagnostic ratios of tracer species were used. The diagnostic ratio is a method to qualitatively identify pollution sources based on the ratios of different tracer species in different emission sources [9,10]. At present, there has been a lot of research into the ratio of chemical components [12,26,42,43,44]. The ratios of different tracer species are shown in Appendix A.

#### 2.4.3. Step 3: Cancer and Non-Cancer Risk Assessment for Different Directions

To evaluate the non-cancer (hazard quotient (*HQ*)) and incremental lifetime cancer risk (*ILCR*) for different directions, HMs and PAHs are the main hazards to be considered, based on the United States Environmental Protection Agency (USEPA) risk assessment model [45,46,47,48]. In general, the *HQ* was estimated based on BaP, Mn, Cr, Ni, As, and V, and the *ILCR* was estimated based on BaA, BbF, BKF, BaP, IPY, DBA, Pb, Cr, Ni, and As. We mainly discuss the effects of respiratory exposure in this study [4,17,21]. The chronic inhalation intake (*CDI*), *HQ*, and *ILCR* of the *j*th clusters were calculated according to Equations (1)–(5), respectively:*CDI_ij_* = *C_ij_ × InhR* × *EF* × *ED* × 10^−6^/*BW*/*AT*(1)
where *C_ij_* is the concentration of *i*th PAHs or *i*th HMs (ug m^−3^) of the *j*th clusters calculated by Step 1 or the concentration of *i*th PAHs or *i*th HMs (ug m^−3^) of the *j*th season. The air inhalation rate, *InhR*, is 20 m^3^ per day^−1^ [45,46,47,48]. *EF* is the exposure frequency (days years^−1^), which was assumed to be 182 days years^−1^ (cold season) and 183 days years^−1^ (warm season). When calculating risks from different directions, the *EF* of every cluster was calculated based on the weight of each direction. The other parameters (*ED*, *BW*, *AT*) used in *CDI* formulas are presented in Appendix A. Cancer risks (*ILCR*) of the PAHs and HMs are calculated based on the inhalation slope factor (*SF*, mg/mg kg^−1^ day^−1^) and the *CDI*, as shown below:*ILC**R_ij_* = *CDI_ij_* × *SF_i_*(2)
*ILCR_j_* = Ʃ*R_ij_*(3)
where *ILC**R_ij_* is the lifetime cancer risk level of carcinogenic element *i* of the cluster *j*. *ILC**R_j_* is presented as the total risk assessment value by summing of *ILC**R_ij_*. *SFi* values of the HMs and PAHs are given in Appendix A. The four thresholds for cancer risks are referenced in this study: acceptable cancer risk level (cancer risk is lower than 1 × 10^−6^), possible cancer risk level (1 × 10^−6^~1 × 10^−5^), probable cancer risk level (1 × 10^−5^~1 × 10^−4^), and definite cancer risk level (> 1 × 10^−4^) [45,46,47].

Non-cancer risks resulting from exposure to PAHs and HMs were evaluated by the reference dose (*RfD*) and hazard quotient (*HQ*), which is defined as follows:*HQ_ij_* = *CDI_ij_/RfD_i_*(4)
*HQ_j_* = Ʃ*HQ_ij_*(5)
where *HQ_ij_* is the non-cancer risk of non-carcinogenic element *i* of the cluster *j*, *HQ_j_* is the sum of non-cancer risks of the cluster *j*, and *RfD_i_* is the reference dose (mg/(kg·day)). *CDI_ij_* is calculated based on Equation (1). *RfD_i_* values of the HMs and PAHs are also shown in Appendix A. When the *HQ* < 1, noncancerous effects are unlikely to occur. When the *HQ* ≥ 1, adverse health effects on humans may occur. When the *HQ* > 10, humans are at high chronic risk [4,17,21].

## 3. Results and Discussion

### 3.1. Characteristics of PM_2.5_ and Component Concentrations

The seasonal average concentrations of PM_2.5_ and its components (water-soluble ions, elements, OC, and EC) are shown in Table 1. The annual PM_2.5_ average concentration was 47 ± 31 μg m^−3^, and 58% of the days in a year exceed the Second Grade National Standard (annual average value: 35 μg/m^3^ for PM_2.5_) established by the Ministry of Environmental Protection of China (GB 3095-2012). The average concentrations of PM_2.5_ were higher in the cold season (58 ± 35 μg m^−3^) than in the warm season (41 ± 21 μg m^−3^). These high PMs, SNA and element levels in the cold season could be related to emission sources, such as coal combustion for heating systems [10,11]. Secondly, the diffusion conditions in the cold season are also weaker than those in the warm season (Appendix A). The average concentration of SNA (SO_4_^2−^ + NO_3_^−^ + NH_4_^+^) and the average concentrations of the elements also were higher in the cold season than in the warm season. Si, Al, Ca, K, Na, and Fe were the most common inorganic elements (Table 1).

The sampling site is close to the Bohai Sea, and the chemical species concentrations may be influenced by sea salt. Based on the Na^+^ concentrations and the seawater compositions, the sea salt components were evaluated [10,49,50]. The percentages of sea salt components accounting for the total amount of the corresponding chemical species in cold and warm seasons are shown in Appendix A. The percentages of ss-Cl^−^ to the total Cl^−^ exceeds 100%, meaning that Cl^−^ loss frequently occurred at Tuoji Island (TI) [50]. The percentages of ss-Mg^2+^ to the total Mg^2+^ also exceeds 100% in many samples, which was similar to Cheju Island, Kosan [50]. The earthly causes are sampling or analytical errors, and some sodium might be from non-marine sources, such as dust. The percentages of ss-K^+^ to total K^+^ and ss- SO_4_^2−^ to total SO_4_^2−^ in PM_2.5_ were all less than 10% (Appendix A), implying that sea salt had little influence on K^+^ and SO_4_^2−^ at TI, even though the sample site was close to the sea [10]. The ss-Ca^2+^ only accounted for 20–40% of the total Ca^2+^ in warm seasons, which indicated that sea salt emissions were a non-negligible source of Ca^2+^ in warm seasons at TI.

The PAH, hopane, sterane, and n-alkane concentrations in cold and warm seasons are presented in Figure 1. The ∑_18_PAHs was higher in the cold season (28.10 ng m^−3^) than in the warm season (5.60 ng m^−3^). These results echo those of Wang et al. [9], who found a higher concentration of ∑15PAHs in the cold-months period than in the warm-months period. BbF and DBA were the most abundant PAHs (17–35% of total PAHs) in the cold and warm seasons. Moreover, these observed values of ∑PAHs are much lower than reported values in urban and rural areas in North China [9,11,49]. Hopanes and steranes are regarded as markers of fossil fuel combustion because they mainly come from lubricating oil residues. Concentrations of hopanes and steranes were higher in the cold season (13.13 ng m^−3^) than in the warm season (5.78 ng m^−3^). The most abundant hopane was 17α(H),21β(H)-hopane, followed by 17α(H),21β(H)-30-Norhopane (Figure 1). The ∑_29_n-alkanes concentrations in PM_2.5_ in the cold season (728.37 ng m^−3^) were higher than those in the warm season (578.92 ng m^−3^).

### 3.2. Seasonal Variations of Cancer and Non-Cancer Risks

The cancer and non-cancer risks assessment for PAHs and HMs in different seasons were calculated according to Equations (1)–(5). The ∑HQ (hazard quotient, non-cancer risks assessment value) was lower in the warm season (0.51) than in the cold season (0.97), which indicated that noncancerous effects were unlikely to occur during the whole sampling time in TI (Figure 2A). Annual average HQ-BaP (0.57) accounted for approximately 39% of the ∑HQ, which was the component with the greatest non-cancer risk (Figure 2C), followed by Ni (32% of ∑HQ) and Mn (18% of ∑HQ). When it comes to the seasonal changes of HQ, it is worth noting that the dominant components to HQ varied with the seasons (Appendix A). HQ-BaP in the cold season (0.41) contributed 42% of ∑HQ, while it only contributed 7% in the warm season (0.04). These results echo those of Wang et al. (2018), who found that higher TEQ-BaP values occurred in the cold season than in the warm period. In the cold season, the increase of HQ-BaP was closely related to additional emissions due to fuel consumption for household heating [10,11] and weak mixing and dispersion capacity (Appendix A).

Contrary to BaP, HQ-Ni in the cold season (0.20) contributed 21% of ∑HQ, which were lower than that contributed 62% of ∑HQ in the warm season (0.32), meaning that Ni emitted from ship sources in warm months are more significant than in cold months [12,16,44]. The main reason is that the Bohai Sea is an inland sea area with a shallow water depth and less influence from the open sea [51]. The salinity of the sea water is relatively low. Under the influence of the cold Siberian air in the cold season, a large area of sea ice can form, which will impede ships [52]. HQ-As contributed the highest proportion in the cold season (0.16) of ∑HQ, and may have been affected by coal emissions caused by domestic heating in northern Chinese cities in the cold season [4,40].

The ∑ILCR (incremental lifetime cancer risk, cancer risk assessment value) in the cold season (1.5 × 10^−4^) was higher than in the warm season (2.5 × 10^−5^), which showed that it was at the probable cancer risk level (1 × 10^−5^~1 × 10^−4^) [45,46,47] (Figure 2B). HMs (Cr and As) dominated the ILCR. These results were consistent with our previous research, which found a higher ILCR for HMs than for PAHs in urban sites [26,34]. Annual average ILCR-Cr (1.90 × 10^−4^) accounted for approximately 98% of the ∑ILCR (Figure 2D), which could be related to vehicle tailpipe emissions, fossil fuel combustion, and industrial processes [31,53,54,55]. In addition, As (1.7 × 10^−6^, accounting for 0.9% of ∑ILCR) can be used as a marker of coal combustion [31,55], leading to the high ILCR of As and Cr. As for the ILCR of PAHs, BaP (2.9 × 10^−7^, accounting for 48% of ∑ILCR-PAHs) and IPY (1.3 × 10^−7^, 22% of ∑ILCR-PAHs) were the largest contributors to the ILCR of PAHs (Figure 2E), meaning that PAHs produced an acceptable cancer risk level to humans. When it comes to the seasonal changes in ILCR, the ILCR-Cr and ILCR-As values in the cold seasons were higher than in the warm seasons (Appendix A), which could be related to more fossil fuel combustion emissions from the inland direction in cold months. However, ILCR-Ni values in the cold season (2.9 × 10^−7^) were lower than in the warm season (4.6 × 10^−7^), which means that Ni emitted from ships posed a possible cancer risk level to humans in summer [12,16,45]. Studies have shown that the bioaccessibilities of metals in simulated human lung fluids are between 10–80%. The bioavailability of Cr is relatively low, about 20–40% [7,56]. In this study, we did not consider the impact of the bioavailability of different hazard components which would lead to an overestimation of the risk assessment.

### 3.3. Directional Variations of Compositions

To identify the potential directions of air masses at TI, the backward trajectory was calculated during the sampling period. Five groups of air masses were identified (Appendix A and Figure 3). According to the directions and pathways of the air masses, five clusters were divided into two categories (trajectories from inland directions and from marine directions). Cluster 1 (accounting for 17.7% of total trajectories), Cluster 2 (11.8%), Cluster 3 (14.0%), and Cluster 4 (15.0%) were classified as inland direction clusters, while Cluster 5 was identified as a marine direction cluster and accounted for 41.6% of the total trajectories.

To explore the potential sources from different directions, the detailed equations are provided in Appendix A. The diagnostic ratios of tracer species were calculated and are shown in Figure 4a and Appendix A. Usually, V and Ni are considered as tracers of heavy fuel oil combustion [16], and they are employed as typical tracers of ship emissions when V/Ni and V/Pb are higher than 0.7 and 0.27, respectively [12,13]. The values of V/Ni and V/Pb were higher than 0.7 and 0.27 only for the marine direction cluster (Figure 4a, Appendix A), evidently indicating that the marine direction cluster was influenced by shipping emissions. The value of Cu/Zn was the lowest for the marine direction cluster (Figure 4a), showing that the marine direction cluster also included diesel combustion [57]. The lowest C29αβ/C30αβ value for the marine direction cluster can also be taken to indicate heavy fuel oil and diesel combustion [17]. Overall, the marine direction cluster originating from the Bohai Sea was obviously influenced by shipping emissions, including heavy oil and diesel ships.

Particulate matter from inland is mostly affected by multiple sources, such as motor vehicles, coal-burning sources, industrial sources, and secondary reactions [23,24,25,31,32]. The sources of PM_2.5_ from inland are more complex than from the sea. The low carbon preference index (CPI, <2) values over all clusters (Figure 4b) implied their anthropogenic impacts [43,58]. The higher OC/EC ratio in inland direction clusters compared to the marine direction cluster indicated the stronger influence of coal combustion, industry, vehicles emissions, or secondary organic aerosols [17,23,44]. The Cu/Zn value was the highest for Cluster 4 (inland direction cluster), indicating its strong association with gasoline vehicles. IcdP/(IcdP + BghiP) and Flt/(Flt + Pyr) for Cluster 4 were 0.22 and 0.46 (Figure 4b, Appendix A), which also indicates that gasoline vehicles play important roles in this cluster [11,59]. This was consistent with the relatively large number of motor vehicles in the Beijing–Tianjin–Hebei (BTH) region [60]. BaP degrades faster than BeP, which may be strongly influenced by photodegradation and long-distance transport [1,2,11]. In this study, the lowest and highest BaP/BeP values were for Cluster 2 (inland direction cluster) and Cluster 4 (Figure 4b), respectively. This suggests that PAHs carried from the BTH region were fresher than those from Inner Mongolia.

When comparing the compositions in marine and inland direction clusters, it was found that the percentages of ss-Cl^−^, ss-Mg^2+^, ss-K^+^, ss- SO_4_^2−^, and ss-Ca^2+^ to total Cl^−^, Mg^2+^, K^+^, SO_4_^2−^, and Ca^2+^ concentrations were higher in the marine direction cluster than in the inland direction clusters (Appendix A), meaning that the marine direction cluster was more effected by sea salt emission at TI. The C34αβS/C34αβS + C34αβR value for hopanes (Figure 4b and Appendix A) was lower in the marine direction cluster (0.24) than in the inland direction clusters (0.51), meaning that air masses from the marine direction cluster were influenced by low maturity fuel combustion. Lower values of C29/C17 and terrigenous-to-aquatic ratios (TARs) implied increased contributions from aquatic inputs [43]. The values of TAR and C29/C27 of n-alkanes were lower in the marine direction cluster (0.15 and 0.27) than in the inland direction clusters. This observation suggested that marine air masses were more influenced by aquatic macrophytes and plankton than inland masses. Moreover, the chemical compositions among inland and marine masses and between the different clusters exhibit significant differences (Student’s *t*-test *p*-values < 0.05). Finally, we come to the conclusion that using the diagnostic rate of organic and inorganic tracers, this study can roughly identify the sources of the marine direction cluster and the inland direction cluster.

### 3.4. Directional Variations of Risks

Cancer and non-cancer risks of PAHs and HMs of different clusters from inland and marine direction clusters were calculated according to Equations (1)–(5). The HQ value of the marine direction cluster (0.59) was below the acceptable limit of non-cancer risk (HQ = 1) (Figure 5). However, the sum of the HQ value for the inland direction clusters (1.16) exceeded the acceptable limit of non-cancer risk (HQ = 1) (Figure 5). The HQ of Cluster 4 was higher than other inland direction clusters. HQ-BaP (0.47) dominated the non-cancer risk for Cluster 4, contributing approximately 82% of HQ_cluster4_. Studies show that BaP is associated with fossil fuel combustion [17]. The HQ of Ni (0.08), BaP (0.05), and Mn (0.05) dominated the non-cancer risk of Cluster 1. Mn is associated with industrial emissions, coal combustion, and the brake and machine wear of road vehicles [4,21,61]. Ni could come from several sources, such as oil combustion, buses, trucks, thermal power plants, incineration, and the oil industry [13,58]. Thus, fossil fuel combustion, industry, and vehicle emissions played an important role in increasing HQ_cluster1_. The HQ of Mn (0.05), Ni (0.04), and As (0.04) dominated the HQ_cluster2_ (0.14). Coal combustion is a major source of As [17,56]. Thus, vehicle emissions, industry, and coal combustion contributed a certain non-cancer risk in Cluster 2. The HQ of As (0.06), Mn (0.05), Ni (0.05), and BaP (0.04) dominated the contributions to the non-cancer risk of Cluster 3 (0.22). Coal combustion, industry, and vehicle emissions played an important role in increasing HQ_cluster3_.

HQ-HMs (0.56) dominated the HQ of the marine direction cluster (0.59), and the HQ-Ni (0.30) dominated the HQ of HMs of the marine direction cluster. What is more, the HQ-Ni and HQ-V of the marine direction cluster were much higher than those of the other 4 inland direction clusters (Figure 5). The HQ-Ni of the marine direction cluster was 1.4 times that of the sum HQ-Ni of the inland direction clusters. The HQ-V of the marine direction cluster was 1.8 times that of the sum HQ-V of inland direction clusters. Usually, V and Ni are considered as tracers of heavy fuel oil combustion by shipping [16]. Thus, ship emissions from the marine direction posed a certain non-cancer risk.

The sum of ILCR values for inland direction clusters (9.3 × 10^−5^) and the marine direction cluster (6.0 × 10^−5^) were at the probable cancer risk level (1 × 10^−5^~1 × 10^−4^) (Figure 5). The ILCR values of inland direction Cluster 1 (2.6 × 10^−5^), Cluster 2 (2.6 × 10^−5^), Cluster 3 (1.8 × 10^−5^), and Cluster 4 (2.3 × 10^−5^) were also at the probable cancer risk level (Figure 5). The air particles from inland and marine directions aggravated the carcinogenic toxicity of the particles. ILCR-Cr (1.8 × 10^−5^~5.5 × 10^−5^) dominated the cancer risk in all clusters, contributing approximately 92–96%. This is mainly because Cr is more carcinogenic than the other components. The ILCR of As in inland direction Cluster 1 (1.2 × 10^−6^), Cluster 2 (1.1 × 10^−6^), and the marine direction cluster (3.7 × 10^−6^) exceeded the possible cancer risk level (1 × 10^−6^~1 × 10^−5^). ILCR-Ni in the marine direction cluster (4.3 × 10^−7^) was much higher than that of other inland direction clusters, while the ILCR-Ni in other inland direction clusters can be ignored (ILCR was much lower than 1 × 10^−6^). Thus, the cancer risk of Ni from ship emissions in the Bohai Sea needs to be paid attention to. The ILCR of PAHs in all clusters did not exceed the acceptable cancer risk level (1 × 10^−6^). The ILCR of Cluster 4 was the highest among all clusters; IPY (4.2 × 10^−7^) and BaP (2.8 × 10^−8^) dominated the ILCR of PAHs in Cluster 4. This result is consistent with the results reported by Zhang et al. [11], who found that rich, high molecular-weight PAHs were mainly contributed by source emissions in the BTH region. Thus, the higher ILCR of PAHs in Cluster 4 was mainly due to vehicle emissions and coal combustion.

## 4. Conclusions

The potential geographical origins of HM and PAH risks and the possible sources from individual directions at Tuoji Island from January 2017 to October 2017 were explored. Concentrations of PM_2.5_ and the most common species, including ∑_18_PAHs, ∑hopanes and steranes, and ∑_29_n-alkanes, were higher in the cold season than in the warm season. The ∑HQ and ∑ILCR were lower in the warm season than in the cold season. HQ-BaP, HQ-Ni, and HQ-As dominated the ∑HQ in cold season, while the non-carcinogenic risk for the warm season was mainly dominated by HQ-Ni, Mn, and As. ILCR-Cr and ILCR-As values were higher in the cold season, while ILCR-Ni was higher in the warm season than in the cold season.

Ship emissions and sea salt were the main sources of atmospheric particulate matter from marine directions, while coal combustion, vehicle emissions, industrial process, or secondary organic aerosols were the main source categories for the inland direction. The non-cancerous effects of TI were mainly affected by inland transport, especially from the inland transport directions of the southeast, northwest, and west-northwest. The cancerous effects of TI were mainly simultaneously affected from inland and marine direction transport. The sum of non-cancer risks from inland directions exceeded the acceptable limit of non-cancer risk. The particles from inland directions aggravated the carcinogenic toxicity of the particles. The sum of non-cancer risks from marine directions did not exceed the acceptable limit of non-cancer risk, while marine sources could contribute a possible cancer risk. It is necessary to pay more attention to the impact of ship exhaust emissions on atmospheric particulate matter, strengthen ship management in ports, and promote the use of clean fuels by ships.

## Figures and Tables

**Figure 1 toxics-10-00032-f001:**
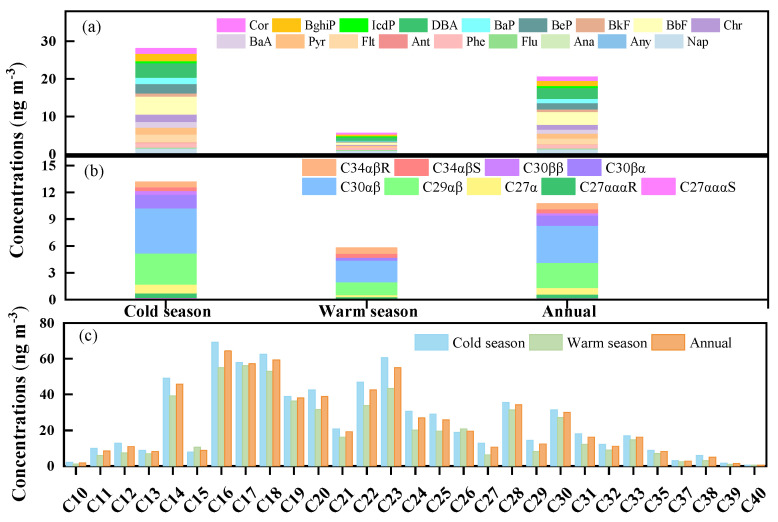
Warm and cold seasonal variation and annual average concentrations of PAHs (**a**), hopanes and sterane (**b**), and n-alkanes (**c**) in PM_2.5_.

**Figure 2 toxics-10-00032-f002:**
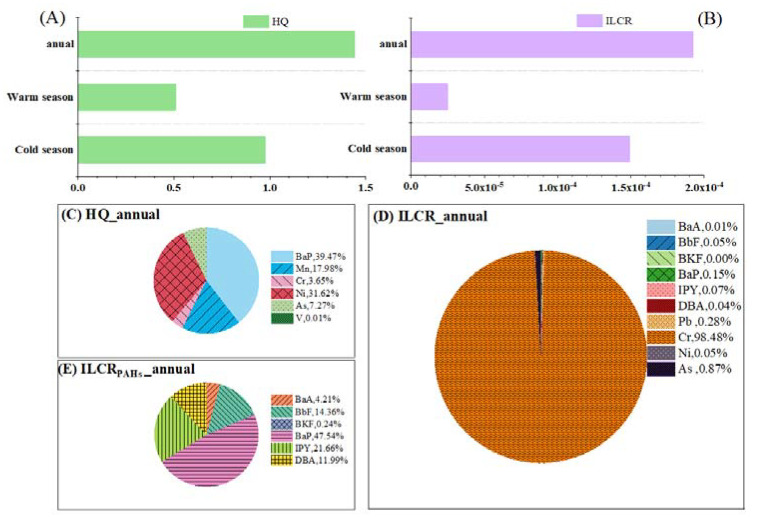
(**A**) HQs of different seasons. (**B**) ILCRs of different seasons. (**C**) The annual proportion of the HQ of each component (PAHs and HMs) relative to the total HQ. (**D**) The annual proportion of the ILCR of each component (PAHs and HMs) relative to the total ILCR. (**E**) The annual proportion of the ILCR of each PAH relative to the total ILCR.

**Figure 3 toxics-10-00032-f003:**
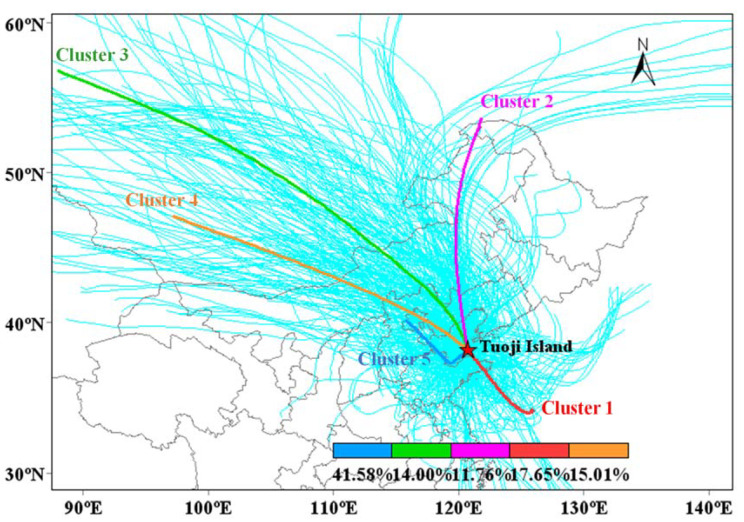
Location of the sampling site (red solid star) and the backward trajectory clusters during the sampling period. Backward trajectory analysis in 24 h started at 500 m altitude and was computed by the NOAA HYSPLIT model. The percentages denote the frequencies of total trajectories.

**Figure 4 toxics-10-00032-f004:**
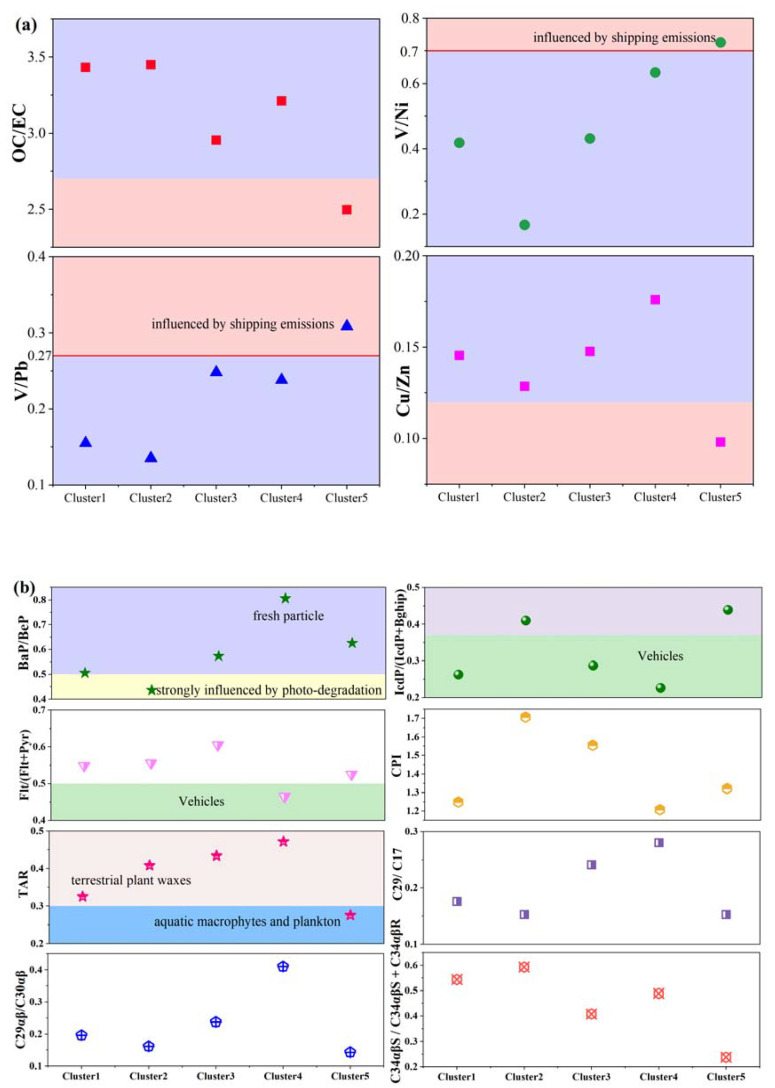
Diagnostic ratios of (**a**) OC/EC, V/Ni, V/Pb, and Cu/Zn for the five clusters, (**b**) the CPI, the PAH congener diagnostic ratios (Flt/(Flt + Pyr), BaP/BeP, IcdP/(IcdP + Bghip)), the ratio of C29/C17 and the terrigenous to aquatic ratio (TAR), and the ratio of HP29/HP30 and 22S/22S + 22R for the five clusters.

**Figure 5 toxics-10-00032-f005:**
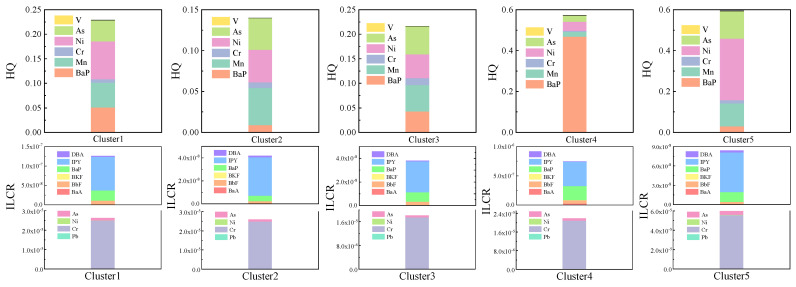
The HQ (non-cancer risk) and ILCR (cancer risk) for each cluster from diverse regions to PM_2.5_ in TI.

**Table 1 toxics-10-00032-t001:** Concentrations of the particle mass (μg m^−3^) and components (μg m^−3^) in PM_2.5_ for warm and cold seasons and annually.

Species	Cold Season (*n* = 32)		Warm Season (*n* = 23)		Annual (*n* = 55)	
	Mean	SD	Mean	SD	Mean	SD
PM_2.5_	58	35	41	21	51	31
Si	2.18	1.85	0.86	0.54	1.65	1.33
Al	0.16	0.10	0.15	0.10	0.16	0.1
Ca	0.60	0.45	0.50	0.23	0.56	0.39
K	0.71	0.47	1.10	1.08	0.87	0.8
Fe	0.33	0.20	0.19	0.11	0.27	0.18
Zn	0.07	0.06	0.03	0.02	0.05	0.04
Na	0.55	0.35	1.29	0.80	0.85	0.68
Mg	0.08	0.05	0.12	0.07	0.09	0.06
Cu	9.15 × 10^−3^	7.89 × 10^−3^	1.30 × 10^−3^	1.28 × 10^−3^	4.50 × 10^−3^	8.90 × 10^−4^
Ba	3.42 × 10^−3^	2.36 × 10^−3^	4.35 × 10^−3^	2.95 × 10^−3^	3.80 × 10^−3^	2.60 × 10^−3^
Ti	2.51 × 10^−2^	1.93 × 10^−2^	5.58 × 10^−3^	2.99 × 10^−3^	1.70 × 10^−2^	1.60 × 10^−2^
V	4.78 × 10^−3^	4.73 × 10^−3^	1.56 × 10^−3^	1.28 × 10^−3^	3.50 × 10^−3^	3.20 × 10^−3^
Mn	1.64 × 10^−2^	1.33 × 10^−2^	7.92 × 10^−3^	3.92 × 10^−3^	1.30 × 10^−2^	1.10 × 10^−2^
Cr	5.71 × 10^−2^	4.07 × 10^−2^	7.05 × 10^−3^	6.54 × 10^−3^	3.70 × 10^−2^	3.60 × 10^−2^
Ni	5.64 × 10^−3^	4.01 × 10^−3^	7.49 × 10^−3^	3.95 × 10^−3^	6.40 × 10^−3^	6.30 × 10^−3^
As	4.98 × 10^−3^	4.86 × 10^−3^	1.88 × 10^−3^	1.10 × 10^−3^	3.70 × 10^−3^	3.40 × 10^−3^
Pb	2.17 × 10^−2^	2.16 × 10^−2^	7.41 × 10^−3^	6.90 × 10^−3^	1.60 × 10^−2^	1.50 × 10^−2^
OC	6.86	3.96	5.46	3.17	6.3	3.76
EC	2.30	1.62	1.41	0.70	1.94	1.41
NO_3_^−^	10.11	9.45	10.18	9.96	10.14	9.76
SO_4_^2−^	7.10	6.22	5.84	4.70	6.59	6.48
NH_4_^+^	5.40	4.79	5.35	4.27	5.38	4.63
Cl^−^	0.85	0.90	0.55	0.48	0.73	0.71
Na^+^	0.44	0.18	1.07	1.04	0.69	0.64
Mg^2+^	0.10	0.07	0.04	0.03	0.07	0.05
Ca^2+^	0.55	0.51	0.13	0.07	0.38	0.3
K^+^	0.62	0.40	0.93	0.84	0.74	0.64

## Data Availability

Not applicable.

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
