# Peer review of "Potential Risks of PM2.5-Bound Polycyclic Aromatic Hydrocarbons and Heavy Metals from Inland and Marine Directions for a Marine Background Site in North China"

_toxics, 2022, doi:10.3390/toxics10010032_

Round 1

Reviewer 1 Report

The study analyze the impacts of ambient PM2.5-bound ions, OC, EC, heavy metals (HMs), 18 polycyclic aromatic hydrocarbons (PAHs), 7 hopanes, and 29 n-alkanes detected at a marine back-ground atmospheric monitoring station (Tuoji island, China). Also, the different sources depending on air direction are presented in the paper.

Despite the very interesting results of article, the manuscript needs some improvements. Overall, the topic and content of the paper are interesting. I therefore recommend publication of this paper in Toxics after major revision. My comments are included in the PDF attached.

Reviewer 2 Report

The article is well written and the presented results fully address the goals of the study as they are described in the last paragraph of the introduction. I believe that the performed analysis of the data will be helpful to other researchers working in source apportionment in remote sites. Therefore I favor the publication of the manuscript. As minor comments I suggest:   

  1. A) Referring to cluster 1, cluster 2 etc. in the abstract is not helpful for the reader, since there is no mention which trajectories correspond to each cluster. Authors could find a better way to present the effect of each cluster to cancer risk level in the abstract session.
  2. B) In section 2.1.2 the authors state that the research was conducted between January 1 and October 30 (about 300 days). Yet 53 samples were collected (for 1/6 of the days) and it is not clarified how many samples were in the cold and warm periods respectively. I suggest to include a table in the supplementary which will provide information on how many samples were collected in each month of the studied period.

Reviewer 3 Report

The manuscript deals with PM2.5 detected at Tuoji island, the only marine background atmospheric monitoring station in North China. 

The topic fits the aims and scope of the Toxics. In my point of view, manuscript needs in minor transformation. 

1) I do not see formula how Cancer risks (ILCR) was calculated.

2) Did authors adapt  BW (Body weigh) in Eq (1) for China?

Reviewer 4 Report

I leave it to the editors to decide whether this work is sufficiently relevant when applying a model with previously published data. I find the paper presented to us interesting, but I consider it to be a compilation of previous works that the authors have already published in different internationally prestigious journals. They apply a model “the back trajectory model”. In this paper there are no new data, all have been previously published.
Here they mix hydrocarbons, PHAs, with heavy metals, HMs, soluble elements and organic and total carbon, which they have previously dealt with in independent papers. Here they already join everything and make a compilation with this model. Some considerations that I would like to make. I think it is necessary to provide climatic data because the variation in pollution in two different seasons is studied: warm and cold and it would be necessary to know what the temperature and precipitation are, it would also be important to know the wind data, since we are talking about particles that are in the air. I think the introduction contains more than enough information. It is very adequate and has an extensive and up-to-date review. Regarding the sampling design. A single sampling point, with only a few coordinates seems scarce and insufficient, it should have been sampled somewhere else. On line 105, reference 21 is repeated.

Round 2

Reviewer 1 Report

The authors have properly addressed all my comments and suggestions. The manuscript reads really well, and in my opinion, it deserves publication in Toxics.
Some corrections along all paper are necessary in proofs, e.g. sea-son in abstract.

Author Response

Response: Thank you very much for your valuable and helpful comments. The minor errors in the manuscript have been corrected, the “season” and “sea-sons” in the abstract have been collectively expressed as “seasons”.

Reviewer 5 Report

The authors did not adequately answer some questions/observations. Please see these points in the attached file.
